# Neural Epitome Search for Architecture-Agnostic Network Compression

**Daquan Zhou[1], Xiaojie Jin[2]\*, Qibin Hou[1], Kaixin Wang[1], Jianchao Yang[2], Jiashi Feng[1]**

[1]Department of Electrical and Computer Engineering, National University of Singapore
[2]Bytedance Inc., Mountain View, USA
{e0357894,kaixin.wang}@u.nus.edu
andrewhoux@gmail.com
{jinxiaojie,yangjianchao}@bytedance.com
elefjia@nus.edu.sg

## Abstract

Traditional compression methods including network pruning, quantization, low rank factorization and knowledge distillation all assume that network architectures and parameters are one-to-one mapped. In this work, we propose a new perspective on network compression, i.e., network parameters can be disentangled from the architectures. From this viewpoint, we present the Neural Epitome Search (NES), a new neural network compression approach that learns to find compact yet expressive epitomes for weight parameters of a specified network architecture end-to-end. The complete network to compress can be generated from the learned epitome via a novel transformation method that adaptively transforms the epitomes to match weight shapes of the given architecture. Compared with existing compression methods, NES allows the weight tensors to be independent of the architecture design and hence can achieve a good trade-off between model compression rate and performance given a specific model size constraint. Experiments demonstrate that, on ImageNet, when taking MobileNetV2 as backbone, our approach improves the full-model baseline by 1.47% in top-1 accuracy with 25% MAdd reduction, and with the same compression ratio, improves AutoML for Model Compression (AMC) by 2.5% in top-1 accuracy. Moreover, taking EfficientNet-B0 as baseline, our NES yields an improvement of 1.2% but has 10% less MAdd. In particular, our method achieves a new state-of-the-art results of 77.5% under mobile settings (<350M MAdd). Code can be found at https://github.com/zhoudaquan/NES.

## 1 Introduction

Despite the remarkable performance achieved in many applications, powerful deep convolutional neural networks (CNNs) typically suffer from high complexity (Han et al., 2015a). The large model size and computation cost hinders their deployment on resource limited devices, such as mobile phones. Very recently, huge efforts have been made to compress powerful CNNs. Existing compression techniques can be generally categorized into four categories: network pruning (Han et al., 2015a; Collins & Kohli, 2014; Han et al., 2015b), low rank factorization (Jaderberg et al., 2014), quantization (Jacob et al., 2018; Hubara et al., 2017; Rastegari et al., 2016), and knowledge distillation (Hinton et al., 2015; Papernot et al., 2016). Network pruning targets on removing unimportant connections or weights to reduce the number of parameters and multiply-adds (MAdd). Low rank factorization decomposes an existing layer into lower-rank and smaller layers to reduce the computation cost. Weights quantization aims to use less number of bits to store the weights and activation maps. Knowledge distillation uses a well trained teacher network to train a lightweight student network. All of those compression methods assume that the model parameters (weight tensors) must have one-to-one correspondence to the architectures. As a result, they suffer from performance drop since changing architectures will inevitably lead to loss of informative parameters.

---

*correspondence

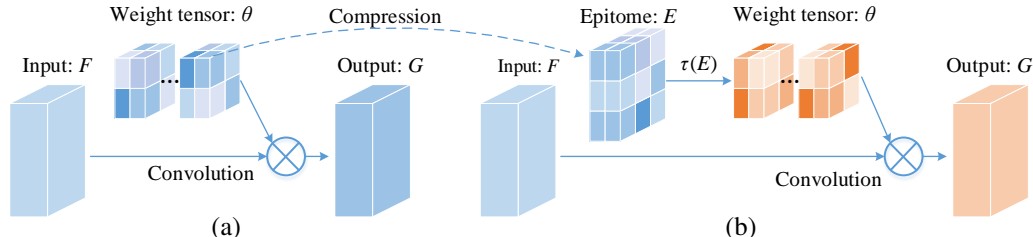

Figure 1: (a) illustrates the conventional convolution process; (b) shows the convolution with NES method. Epitome $E$ has a different shape as defined by the architecture. A transformation is learned automatically to transform the $E$ to a shape that match the architecture defined shape. By designing a smaller $E$, significant compression can be achieved with less performance drop. In certain cases, the performance can be increased with less computation as shown in Table 3.

In this paper, we consider the network compression problem from a new perspective where the shape of the weight tensors and the architecture are designed independently. The key insight is that the network parameters can be disentangled from the architecture and can be compactly represented by a small-sized parameter set (called epitome), inspired by success of epitome methods in image/video modeling and data sparse coding (Jojic et al., 2003; Cheung et al., 2008; Aharon & Elad, 2008). As shown in Figure 1, unlike conventional convolutional layers that use the architecture tied weight tensors to convolve with the input feature map, our proposed neural epitome search (NES) approach first learns a compact yet expressive epitome along with an adaptive transformation function to expand the epitomes. The transformation function is able to generate a variety of parameters from epitomes via a novel learnable transformation function, which also guarantees the representation capacity of the resulting weight tensors to be large. Our transformation function is differentiable and hence enables the NES approach to search for optimal epitome end-to-end, achieving a good trade-off between required model size and performance. In addition, we propose a novel routing map to record the index mapping used for the transformation between the epitome and the convolution kernel. During inference, this routing map enables the model to reuse computations when the expanded weight tensors are formed based on the same set of elements in the epitomes and therefore effectively reduces the computation cost.

Benefiting from the learned epitomic network parameters and transformation method, compared to existing compression approaches, NES has less performance drop. To the best of our knowledge, this is the first work that automatically learns compact epitomes of network parameters and the corresponding transformation function for network compression. To sum up, our work offers the following attractive properties:

- Our method is flexible. It allows the weight tensors to be independent of the architecture design. We can easily control the model size by defining the size of the epitomes given a specified network architecture. This is especially beneficial in the context of edge devices.

- Our method is effective. The learning-based transformation method empowers the epitomes with highly expressive capability and hence incurs less performance drop even with large compression ratios.

- Our method is easy to use. It can be encapsulated as a drop in replacement to the current convolutional operator. There is no dependence on specialized platforms/frameworks for NES to conduct compression.

To demonstrate the efficacy of the proposed approach, We conduct extensive experiments on CIFAR-10 (Krizhevsky & Hinton, 2009) and ImageNet (Deng et al., 2009). On CIFAR-10 dataset, our method outperforms the baseline model by 1.3%. On ImageNet, our method outperforms MobileNetV2 full model by 1.47% in top-1 accuracy with 25% MAdd reduction, and MobileNetV2-0.35 baseline by 3.78%. Regarding MobileNetV2-0.7 backbone, our method improves AMC (He et al., 2018) by 2.47%. Additionally, when taking EfficientNet-b0 (Tan & Le, 2019) as baseline, we have an improvement of 1.2% top-1 accuracy with 10% MAdd reduction.

## 2 RELATED WORK

Traditional model compression methods include network pruning (Collins & Kohli, 2014; Han et al., 2015b), low rank factorization (Jaderberg et al., 2014), quantization (Jacob et al., 2018; Hubara et al.,

2017; Rastegari et al., 2016) and knowledge distillation (Hinton et al., 2015; Papernot et al., 2016). For all of those methods, as mentioned in Section 1 extensive expert knowledge and manual efforts are needed and the process might need to be done iteratively and hence is time consuming.

Recently, AutoML based methods have been proposed to reduce the experts efforts for model compression (He et al., 2018; Zoph et al., 2018; Noy et al., 2019; Li et al., 2019) and efficient convolution architecture design (Liu et al., 2018; Wu et al., 2018; Tan et al., 2018). As proposed in AutoML for model compression (AMC (He et al., 2018)), reinforcement learning can be used as an agent to remove redundant layers by adding resource constraints into the rewards function which however is highly time consuming. Later, gradient based search method such as DARTS (Liu et al., 2018) is developed for higher search efficiency over basic building blocks. There are also methods that use AutoML based method to search for efficient architectures directly (Wu et al., 2018; He et al., 2018). All of those methods are searching for optimized network architecture with an implicit assumption that the weights and the model architecture have one-to-one correspondence. Different from all of the above mentioned methods, our method provides a new search space by separating the model weights from the architecture. The model size can thus be controlled precisely by nature.

Our method is also related to the group-theory based network transformation. Based on the group theory proposed in Cohen & Welling (2016), recent methods try to design a compact filter to reduce the convolution layer computation cost such as WSNet (Jin et al., 2017) and CReLU (Shang et al., 2016). WSNet tries to reduces the model parameters and computations by allowing overlapping between adjacent filters of 1D convolution kernel. This can be seen as a simplified version of our method as the overlapping can be regarded as a fixed rule transformation. CReLu tries to learn diversified features by concatenating ReLU output of original and negated inputs. However, as the rule is fixed, the design of those schemes are application specific and time consuming. Besides, the performance typically suffers since the scheme is not optimized during the training. In contrast, our method requires negligible human efforts and the transformation rule is learned end-to-end.

## 3 METHOD

### 3.1 OVERVIEW

A convolutional layer is composed of a set of learnable weights (or kernels) that are used for feature transformation. The observation of this paper is that the learnable weights in CNNs can be disentangled from the architecture. Inspired by this fact, we provide a new perspective on the network compression problem, i.e., finding a compact yet expressive parameter set, called *epitome*, along with a proper transformation function to fully represent the whole network, as illustrated in Figure 1.

Formally, consider a convolutional network with a fixed architecture consisting of a stack of convolutional layers, each of which is associated with a weight tensor $\theta_i$ ($i$ is layer index). Further, let $\mathcal{L}(X, Y; \theta)$ be the loss function used to train the network, where $X$ and $Y$ are the input data and label respectively and $\theta = \{\theta_i\}$ denotes the parameter set in the network. Our goal is to learn epitomes $E = \{E_i\}$ which have smaller sizes than $\theta$ and the transformation function $\tau = \{\tau_i\}$ to represent the network parameter $\theta$ with the compact epitome as $\tau(E)$. In this way, network compression is achieved. The objective function of neural epitome search (NES) for a given architecture is to learn optimal epitomes $E^*$ and transformation functions $\tau^*$:

$$\{E^*, \tau^*\} = \underset{E, \tau}{\arg\min} \, \mathcal{L}(X, Y; \tau(E)), \qquad s.t. \ |E| < |\theta|, \qquad (1)$$

where $|\cdot|$ calculates the number of all elements.

The above NES approach provides a flexible way to achieve network compression since the epitomes can be defined to be of any size. By learning a proper transformation function, the epitomes of predefined sizes can be adaptively transformed to match and instantiate the specified network architecture. In the following sections, we will elaborate on how to learn the transformation functions $\tau(\cdot)$, the epitome ($E \in \mathbb{R}^{W^E \times H^E \times C_{in}^E \times C_{out}^E}$) and how to compress models via NES in an end-to-end manner. In this paper, we use "a sub-tensor in the epitome" to describe a patch of the epitome that will be selected to construct the convolution weight tensor. The sub-tensor, $E_s$, is represented with the starting index and the length along each dimension in the epitome as shown below:

$$E_s = E[p : p + w, q : q + h, c_{in} : c_{in} + \beta_1, c_{out} : c_{out} + \beta_2], \qquad (2)$$

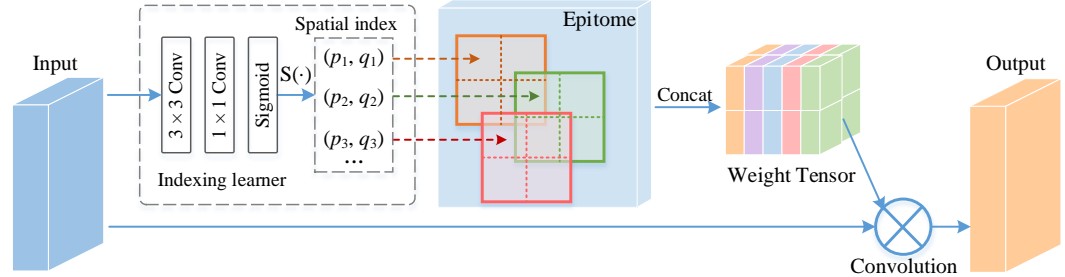

Figure 2: Our proposed compression process along the spatial dimension. We only show the transformation along spatial dimensions for easy understanding and the transformation along the channel dimension can be found in Figure 3. The indexing learner learns the position mapping function, $\mathcal{M} : (i, j, m) \rightarrow (p, q)$, between the convolution kernel elements and the sub-tensor in the epitome $E_s$. The learned starting indices and the epitome are fed into Eqn. (7) to sample the weight tensor. The outputs of Eqn. (7) are concatenated together to form the weight tensor. Note that we use a moving average way to update $\mathcal{M}$ so that the indexing learner can be removed during inference. This is shown in details in section 3.2 in the paragraph 'Routing map'. The whole training process is end-to-end trainable and hence can be used for any specified network architecture. Here, we abuse the notion for the starting index pair $(p_n, q_n)$ by using subscript $n$ to denote the $n^{th}$ pair of the starting index during sampling(Eqn. (7)) while in the main text, we use subscript $t$ to denote the training epoch. The learned indices and the epitome are fed into the interpolation-based sampler (Eqn. (7)) and the outputs of Eqn. (7) are concatenated together to form the convolution weights tensor.

where $(p, q, c_{in}, c_{out})$ denote the starting index of the sub-tensor and $(w, h, \beta_1, \beta_2)$ denotes the length of the sub-tensor along each dimension.

## 3.2 Differentiable Search for Epitome Transformation

As aforementioned, NES generates convolution weight tensors from the epitome $E$ via a transformation function $\tau$. In this section, we explain how the transformation $\tau$ is designed and optimized. We start with the formulation of a conventional convolution operation. Then we introduce how the transformed epitome is deployed to conduct the convolution operations, with a reduced number of parameters and calculations.

A conventional 2D convolutional layer transforms an input feature tensor $F \in \mathbb{R}^{W \times H \times C_{in}}$ to an output feature tensor $G \in \mathbb{R}^{W \times H \times C_{out}}$ through convolutional kernels with weight tensor $\theta \in \mathbb{R}^{w \times h \times C_{in} \times C_{out}}$. Here $(W, H, C_{in}, C_{out})$ denote the width, height, input and output channel numbers of the feature tensor; $w$ and $h$ denote width and height of the convolution kernel. The convolution operation can be formulated as:

$$G_{t_w, t_h, c} = \sum_{i=0}^{w-1} \sum_{j=0}^{h-1} \sum_{m=0}^{C_{in}-1} F_{t_w+i, t_h+j, m} \theta_{i,j,m,c}, \forall \ t_w \in [0, W), t_h \in [0, H), c \in [0, C_{out}). \quad (3)$$

Instead of maintaining the full weight tensor $\theta$, NES maintains a much smaller epitome $E$ that can generate the weight tensor and thus achieves model compression. To make sure the generated weights $\tau(E)$ can conduct the above convolution operation without incurring performance drop, we carefully design the transformation function $\tau$ with following three novel components: (1) a learnable indexing function $\eta$ to determine starting indices of the sub-tensor within the epitome $E$ to sample the weight tensor; (2) a routing map $\mathcal{M}$ that records the location mapping from the sampled sub-tensors epitome to the generated weight tensor; (3) an interpolation-based sampler to perform the sampling from $E$ even when the indices are fractional. We now explain their details.

**Indexing function**  The indexing function $\eta$ is used to localize the sub-tensor within the epitome that is used to generate the weight tensor, as illustrated in Figure 2. Concretely, given the input feature tensor $F \in \mathbb{R}^{W \times H \times C_{in}}$, the function generates indices as follows,

$$(\mathbf{p}, \mathbf{q}, \mathbf{c_{in}}, \mathbf{c_{out}}) = S(\eta(F)), \quad (4)$$

where $\mathbf{p}, \mathbf{q}, \mathbf{c_{in}}, \mathbf{c_{out}}$ are vectors of learned starting indices along the spatial, input channel and filter dimensions respectively to sample the sub-tensor within epitome to generate the model weight tensors.

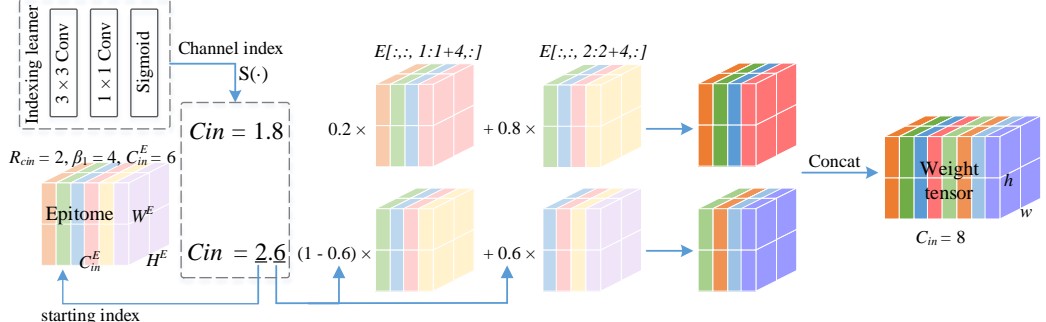

Figure 3: Transformation along the input channel dimension of NES. In the figure, we only show three dimensions of the epitome with $E \in \mathbb{R}^{W^E \times H^E \times C_{in}^E \times 1}$. To simplify the illustration, we set $W^E = w$ and $H^E = h$ where $w$ and $h$ are the size of the convolution kernel. Thus, the starting indices along the spatial dimension,(p,q), are not shown in the figure. The generated weight tensor has input channel number equal to 8. $R_{cin} = \lceil C_{in}/\beta_1 \rceil = 2$ denotes the number of samplings applied along the input channel dimension. During each transformation, sub-tensor with shape $w \times h \times \beta_1$ is selected each time based on Eqn. (8) by replacing the starting index (p,q) in Eqn. (7) with $(c_{in})$ and enumerating over the input channel dimension. In this example, $\beta_1$ is set to 4.

Each vector contains a set of starting indices and the element number inside each vector is equal to the number of transformations that will be applied along each dimension. Note they are all non-negative real numbers. $\eta$ is the index learner and it outputs the normalized indices $(\mathbf{p}', \mathbf{q}', \mathbf{c}'_{\mathbf{in}}, \mathbf{c}'_{\mathbf{out}})$ through a sigmoid function, each ranging from 0 to 1. These outputs are further up-scaled by a scaling function $S(\cdot)$ to the corresponding dimension of the epitome by $S(\cdot)$

$$S(\mathbf{p}', \mathbf{q}', \mathbf{c}'_{\mathbf{in}}, \mathbf{c}'_{\mathbf{out}}) = [W^E, H^E, C_{in}^E, C_{out}^E] \otimes [\mathbf{p}', \mathbf{q}', \mathbf{c}'_{\mathbf{in}}, \mathbf{c}'_{\mathbf{out}}], \qquad (5)$$

where $W^E, H^E, C_{in}^E, C_{out}^E$ are dimensions of the epitome and $\otimes$ denotes element-wise multiplication. The learned indices are then fed into the following interpolation based sampler to generate the weight tensor. We implement the indexing learner by a two-layer convolution module that can be jointly optimized with the backbone network end-to-end. In particular, we use separate indexing learners and epitomes for each layer of the network. More implementation details are given in Appendix A.3.

**Routing map** The routing map is constructed to record the position correspondence between the convolution weight tensor and the sub-tensor in the epitome. It takes a position within the weight tensor as input and returns the corresponding starting index of the sub-tensor in the epitome. The mapped starting index of the sub-tensor in the epitome can thus be retrieved from the routing map fast. More importantly, the indexing learner can be removed during inference with the help of the routing map. The routing map is built as a look-up table during the training phase by recording the moving average of the output index from the index learner $\eta$. For example, the starting index pair as shown in Figure 2 can be fetched via $(p_t, q_t) = \mathcal{M}(i, j, m)$ where $(i, j, m)$ is the spatial location in the weight tensor and $(p_t, q_t)$ is the starting index of the selected sub-tensor in the epitome at training epoch $t$. The routing map $\mathcal{M}$ is constructed via Eqn. (6) as shown below with momentum $\mu$ during the training phase. $\mu$ is treated as a hyper-parameter and is decided empirically[1]:

$$\mathcal{M}(i, j, m) = (p_t, q_t) = (p_{t-1}, q_{t-1}) + \mu \cdot \eta(x). \qquad (6)$$

**Interpolation based sampler** The learned starting index and the pre-defined dimension $(w, h, \beta_1, \beta_2)$ of the sub-tensor in the epitome is then fed into the sampler function. The sampler function samples the sub-tensor within the epitome to generate the weight tensor. To simplify the illustration on the sampler function, we use the transformation along the spatial dimension as an example as shown in Figure 2. The weight tensor is generated via the equation as shown below:

$$\theta_{(:,:,m)} = \tau(E|(p,q)) = \sum_{n_w=0}^{W^E-1} \sum_{n_h=0}^{H^E-1} \mathcal{G}(n_w, p)\mathcal{G}(n_h, q)E_{(n_w:n_w+w, n_h:n_h+h)}, \qquad (7)$$

where $\mathcal{G}(a, b) = \max(0, 1 - |a-b|)$; $n_w$ and $n_h$ enumerate over all integral spatial locations within $E$. Following Eqn. (7), we first find all sub-tensors in the epitome whose starting indices $(n_w, n_h)$ along

---

[1]We set $\mu$ to be 0.97 in our experiments.

spatial dimensions satisfy: $\mathcal{G}(n_w, p_t)\mathcal{G}(n_h, q_t) > 0$. Then a weighted summation (or interpolation) over the involved sub-tensors is computed according to Eqn. (7). In the case of applying the sampling along input channel dimension, $(p, q)$ in the equation is replaced with the learned starting index $c_{in}$ and the weight tensor is generated by iterating along the input channel dimension as shown below:

$$\theta_{(:,:,m:m+\beta_1)} = \tau(E|c_{in}) = \sum_{n_c=0}^{R_{cin}-1} \mathcal{G}(n_w, c_{in})E_{(:,:,c_{in}:c_{in}+\beta_1)}, \tag{8}$$

Where $R_{cin} = \lceil C_{in}/\beta_1 \rceil$ is the number of samplings applied along the input channel dimension. An example of the generation process with Eqn. (8) along the channel dimension can be found in Figure 3.

The transformation function can be applied in any dimension of the weight tensor. Figure 2 illustrates the transformation along the spatial dimension and Figure 3 shows the transformation along the input channel dimension. The transformation along the filter dimension is the same as the transformation along the input channel dimension. However, transformation along the filter dimension is easier for the computation reuse. We will show this in details in section 3.4.

### 3.3 LEARNING TO SEARCH EPITOMES END-TO-END

Benefiting from the differentiable Eqn. (7), the elements in $E$ and the transformation learner $\eta$ can be updated together with the convolutional layers through back propagation in an end-to-end manner. For each element in the epitome $E$, as its transformed weight parameter can be used in multiple positions in weight tensor, the gradients of the epitome are thus the summation of all the positions where the weight parameters are used.

Here, for clarity, we use $\{\tau^{-1}(p,q)\}$ to denote the set of the indices in the convolution kernel that are mapped from the same position $(p, q)$ in $E$. Note that here we abuse the notion of $(p, q)$ to denote the integer spatial position in the epitome. The gradients of $E_{(p,q)}$ can, thus, be calculated as:

$$\nabla_{E_{(p,q)}}\mathcal{L} = \sum_{z \in \{\tau^{-1}(p,q)\}} \alpha_z \nabla_{\theta_z}\mathcal{L}, \tag{9}$$

where $\theta_z$ is the kernel parameters that are transformed from $E_{(p,q)}$, and $\alpha_z$ are the fractions that are assigned to $E_{(p,q)}$ during the transformation. The epitome can thus be updated via Eqn. (10):

$$E^t_{(p_t,q_t)} = E^{t-1}_{(p_t,q_t)} - \epsilon \nabla_{E^{t-1}_{(p_t,q_t)}}\mathcal{L}, \tag{10}$$

where $\epsilon$ denotes the learning rate and subscript $t$ denotes the training epoch. Eqn. (9) and (10) use the parameter updating rule along the spatial dimension as an example. The above equations can be applied on any dimension by replacing the index mapping. The indexing learner $\eta$ can be simply updated according to the chain rule.

### 3.4 COMPRESSION EFFICIENCY

**Parameter reduction** By using the routing map which records location mappings from the sub-tensor in the epitome to the convolution weight tensor, the indexing learner can be removed during the inference phase. Thus, the total number of parameters during inference is decided by the size of the epitome and the routing map. Recall that the epitome $E$ is a four dimensional tensor with shape $(W^E, H^E, C^E_{in}, C^E_{out})$. The size of an sub-tensor in the epitome is denoted as $(w, h, \beta_1, \beta_2)^2$ where $\beta_1 \leq C^E_{in}$ and $\beta_2 \leq C^E_{out}$. The size of the epitome can be calculated as $W^E \times H^E \times C^E_{in} \times C^E_{out}$. The size of the routing map $\mathcal{M}$ is calculated as $3 \times R_{cin} + R_{cout}$ where $R_{cout} = \lceil C_{out}/\beta_2 \rceil$ is the number of starting indices learned along the output channel dimension, and $3 \times R_{cin} = 3 \times \lceil C_{in}/\beta_1 \rceil$ is the number of starting index learned along the spatial and input channel dimension. Note that we can enlarge the size of the sub-tensor in the epitome to reduce the size of the routing map. Here, the size is referring to the number of parameters. Detailed explanations of how $R_{cin}$ is calculated can be found in the Figure 3. The parameter compression ratio $r$ can thus be calculated via Eqn. (11):

$$r = \frac{w \times h \times C_{in} \times C_{out}}{W^E \times H^E \times C^E_{in} \times C^E_{out} + 3 \times R_{cin} + R_{cout}} \approx \frac{C_{out} \times C_{in} \times w \times h}{C^E_{out} \times C^E_{in} \times W^E \times H^E}, \tag{11}$$

---

[2]We set $\beta_1$ to $C^E_{in}$ and $\beta_2$ to $C^E_{out}$ in this paper.

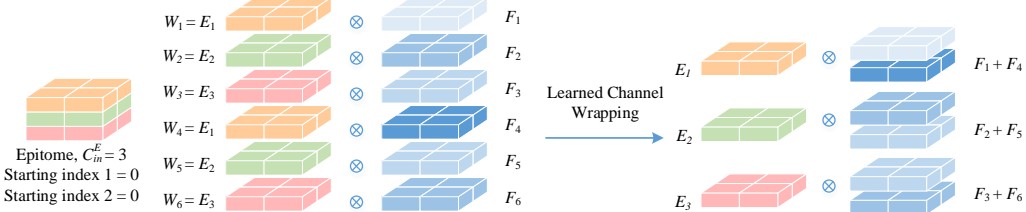

Figure 4: NES transformation along the input channel dimension with channel wrapping. To simplify the illustration, we choose an epitome with shape $\mathbb{R}^{w \times h \times 3 \times 1}$ and $C_{in}^E = 3$. In this example, the transformation is applied twice and the two learned starting indices are 0. The input feature map $\mathcal{F}$ is first added based on the learned interpolation position of the kernels. Input feature map $\mathcal{F}_1$ and $\mathcal{F}_4$ are both multiplied with the first channel in the epitome since $W_1$ and $W_4$ are both generated with $E_1$. To reuse the multiplication, feature map $\mathcal{F}_1$ and $\mathcal{F}_4$ are first added together before multiplying with the weights kernel $E_1$. The figure uses integer index to simplify the illustration. When the learned indices are fractions, the feature maps are the weighted summation of the two nearest integer indexed sub-tensors in the epitome as shown in Eqn. (8). For example, if the two starting indices in this figure are 0.6 and 0.3, the calculation becomes $(0.6F_1 + 0.3F_4 + 0.4F_3 + 0.7F_6) \otimes E1 + (0.4F_1 + 0.7F_4 + 0.6F_2 + 0.3F_5) \otimes E_2 + (0.4F_2 + 0.7F_5 + 0.6F_3 + 0.3F_6) \otimes E_3$. Since we group the feature map first before the convolution, the computation cost is reduced.

From Eqn. (11), it can be seen that the compression ratio is nearly proportional to the ratio between the size of the epitome and the generated weight tensor. Detailed proof can be found in Appendix E. The above analysis demonstrates that NES provides a *precise control of the model size* via the proposed transformation function.

**Computation reduction** As the weight tensor $\theta$ is generated from the epitome $E$, the computation in convolution can be reused when different elements in $\theta$ are from the same portion of elements in $E$.

Concretely, we propose two novel schemes to reuse the computation along the input channel dimension and the filter dimension respectively.

*Channel wrapping.* During the inference, the computation along the input channel dimension is reduced with channel wrapping as illustrated in Figure 4. For the elements in the input feature map that are multiplied with the same element in the epitome, we group the feature map elements first and then multiplied with the weight tensor in the epitome as follows:

$$\tilde{F}(i,j,m) = \sum_{c'=0}^{R_{cin}-1} F(p,q,m + c' \times C_{in}^E + c_{in}),$$
(12)

where $R_{cin} = \lceil C_{in}/C_{in}^E \rceil$ is the number of samplings (Eqn. (8)) applied along the input channel dimension and $m + c_{in} + c' \times C_{in}^E$ is the learned position with $(p, q, c_{in}) = \mathcal{M}(i, j, m)$ and $c_{in} \in [0, C_{in}^E)$. This process is also illustrated in Figure 4.

*Product map and integral map.* For the reuse along the filter dimension, given the routing map $\mathcal{M}$ for the transformation, we first calculate the convolution results between the epitome and the input feature map once and then save the results as a product map $P$.

During inference, given $P$, the multiplication in convolution can be reused in a lookup table manner with $O(1)$ complexity:

$$G_{t_w, t_h, c} = \sum_{i=0}^{W-1} \sum_{j=0}^{H-1} \sum_{m=0}^{C_{in}-1} F_{t_w+i, t_h+j, m} \theta_{\mathcal{M}(i,j,m,c)} = \sum_{i=0}^{W-1} \sum_{j=0}^{H-1} \sum_{m=0}^{C_{in}-1} P_{\mathcal{M}(i,j,m,c)}.$$
(13)

The additions in Eqn. (13) can also be reused via an integral map $I$ (Crow, 1984) as done in Viola et al. (2001). With the product map and the integral map, the MAdd can be calculated as:

$$\text{Reduced MAdd} = (2C_{in}W^E H^E - 1)WHC_{out}^E + WHW^E H^E C_{out}^E + 2R_{cin}WH\beta_1 + 2R_{cout}\beta_2.$$
(14)

Table 1: Comparison with WSNet on ESC-50 dataset. We use the same configuration but change the sampling stride to be learnable. 'S' denotes the stride and 'C' denotes the repetition times along the input channel dimension. We use the 'S' in WSNet as initial values and learns the offsets.

| Method | Conv1 | | Conv2 | | Conv3 | | Conv4 | | Conv5 | | Conv{6-8} | | Acc. (%) | Params |
|--------|---|---|---|---|---|---|---|---|---|---|---|---|---|---|
| Config. | S | C | S | C | S | C | S | C | S | C | S | C | | |
| baseline | 1 | 1 | 1 | 1 | 1 | 1 | 1 | 1 | S | 1 | 1 | 1 | 66.0 | 1× |
| WSNet | 8 | 1 | 4 | 1 | 2 | 2 | 1 | 2 | S | 4 | 1 | 8 | 66.5 | 4× |
| Ours | 8 | 1 | 4 | 1 | 2 | 2 | 1 | 2 | S | 4 | 1 | 8 | **73.0** | 4× |

Table 2: Results of ImageNet classification. Our method uses vanilla MobileVetV2 as backbone. For a fair comparison, we evaluate multiple width multiplier values of 0.75, 0.5, 0.35 and 0.18 and only apply it on the filter dimension of the first $1 \times 1$ convolution. We apply the proposed method on all the invert residual blocks equally to disentangle the architecture affects on the performance. MAdd are calculated based on all convolution blocks with an assumption that the batch normalization layers are merged. '$*$' denotes our own implementation.

| Methods | MAdd(M) | Parameters | Param Compression Rate | Top-1 Accuracy(%) |
|---------|---------|------------|------------------------|-------------------|
| MobilenetV2-1.0 | 301 | 3.4M | 1.00× | 71.8 |
| MobilenetV2-0.75$^*$ | 217 | 2.94M | 1.17× | 69.14 |
| MobilenetV2-0.5$^*$ | 153 | 2.52M | 1.36× | 67.22 |
| MobilenetV2-0.35$^*$ | 115 | 2.26M | 1.54× | 65.18 |
| MobilenetV2-0.18$^*$ | 71 | 1.98M | 1.80× | 60.70 |
| Our method-0.75 | 220 | 2.94M | 1.17× | 71.54 |
| Our method-0.5 | 157 | 2.52M | 1.36× | 69.42 |
| Our method-0.35 | 120 | 2.26M | 1.54× | 67.01 |
| Our method-0.18 | 79 | 1.95M | 1.80× | 64.48 |

Hence, the computation cost reduction ratio can be written as:

$$\text{MAdd Reduction Ratio} = \frac{C_{out}HW(2C_{in}wh - 1)}{C_{out}^E HW(W^E H^E + 2C_{in}^E W^E H^E - 1) + 2R_{cin}WH\beta_1 + 2\beta_2 R_{cout}} \tag{15}$$

See more details and analysis in Appendix E.

**Discussion.** We make a few remarks on the advantages of our proposed method as follows. The proposed NES method disentangles the weight tensors from the architecture by using a learnable transformation function. This provides a new research direction for model compression by bringing in better design flexibility against the traditional compression methods on both sides of software and hardware. On the software side, NES does not require re-implementation of acceleration algorithms. All the operations employed by NES are compatible with popular neural network libraries and can be encapsulated as a drop in operator. On the hardware side, the memory allocation of NES is more flexible by allowing easily adjust the epitome size. This is especially helpful for hardware platform where the off chip memory access is the main power consumption as demonstrated in Han et al. (2016). NES provides a way to balance the computation/memory-access ratio in hardware: a smaller epitome with a complex transformation function results in a computation intense model while a large epitome with simple transformation function results in a memory intensive model. Such ratio is an important hardware optimization criteria which however is not covered by most previous compression methods.

## 4 EXPERIMENTS

We first evaluate the efficacy of our method in 1D convolutional model compression on the sound dataset ESC-50 (Piczak, 2015) for the comparison with WSNet. We then test our method with MobileNetV2 and EfficientNet as the backbone on 2D convolutions on ImageNet dataset (Deng et al., 2009) and CIFAR-10 dataset (Krizhevsky & Hinton, 2009). Detailed experiments settings can be found in Appendix A. For all experiments, we do not use additional training tricks including the squeeze-and-excitation module (Hu et al., 2018) and the Swish activation function (Ramachandran et al., 2017) which can further improve the results unless those are used in the bachbone model

Table 3: Comparison of our method with other state-of-the-art models on ImageNet where our method shows superior performance over all other methods. MAdd are calculated based on all convolution blocks with an assumption that the batch normalization layers are merged. Suffix '-A' means we use larger compression ratio for front layers. Our method does not modify the backbone model architecture and applies a uniform compression ratio, unless specified with suffix '-A'. All experiments are using MobileNetV2 as backbone unless labeled with EfficientNet as suffix.

| GROUP | Methods | MAdd (M) | Params | Top-1 Acc. (%) |
|---|---|---|---|---|
| 60M MAdd | MobilenetV2-0.35 (Sandler et al., 2018) | 59 | 1.7M | 60.3 |
| | S-MobilenetV2-0.35 (Yu et al., 2018) | 59 | 3.6M | 59.7 |
| | US-MobilenetV2-0.35 (Yu & Huang, 2019b) | 59 | 3.6M | 62.3 |
| | MnasNet-A1 (0.35x) (Tan et al., 2018) | 63 | 1.7M | 62.4 |
| | Our method-0.18 | 79 | 2.0M | **64.48** |
| 100M MAdd | MobilenetV2-0.5 (Sandler et al., 2018) | 97 | 2.0 M | 65.4 |
| | S-MobilenetV2-0.5 (Yu et al., 2018) | 97 | 3.6M | 64.4 |
| | US-MobilenetV2-0.5 (Yu & Huang, 2019b) | 97 | 3.6M | 65.1 |
| | Our method-0.35 | 120 | 2.2M | **67.01** |
| 200M+ MAdd | MobilenetV2-0.75 (Sandler et al., 2018) | 209 | 2.6 M | 69.8 |
| | S-MobilenetV2-0.75 (Yu & Huang, 2019a) | 209 | 3.6M | 68.9 |
| | US-MobilenetV2-0.75 (Yu & Huang, 2019b) | 209 | 3.6M | 69.6 |
| | FBNet-A (Wu et al., 2018) | 246 | 4.3M | 73 |
| | AUTO-S-MobilenetV2-0.75 (Yu & Huang, 2019a) | 207 | 4.1M | 73 |
| | Our method-0.5 | **157** | **2.5M** | **69.42** |
| | Our method-0.75 | **220** | **2.9M** | **71.54** |
| | Our method-0.75-A | **225** | **3.7M** | **73.27** |
| | Our method-0.5 (EfficientNet-b0) | **240** | **3.92M** | **75.55** |
| | Our method-0.5 (EfficientNet-b1) | **350** | **5.46M** | **77.5** |

originally. The calculation of MAdd is performed for all convolution blocks. We evaluate our methods in terms of three criteria: model size, multiply-adds(MAdd) and the classification performance.

## 4.1 1D CNN COMPRESSION

For 1D convolution compression, we compare with WSNet. Similar to WSNet (Jin et al., 2017), we use the same 8-layer CNN model for a fair comparison. The compression ratio in WSNet is decided by the stride ($S$) and the repetition times along the channel dimension ($C$), as shown in Table 1. From Table 1, one can see that with the same compression ratio, our method outperforms WSNet by 6.5% in classification accuracy. This is because our method is able to learn proper weights and learn a transformation rules that are adaptive to the dataset of interest and thus overcome the limitation of WSNet where the sampling stride is fixed. More results can be found in Appendix B.

## 4.2 2D CNN COMPRESSION

**Implementation details.** We use both MobilenetV2 (Sandler et al., 2018) and EfficientNet (Tan & Le, 2019) as our backbones to evaluate our approach on 2D convolutions. Both models are the most representative mobile networks very recently and have achieved great performance on ImageNet and CIFAR-10 datasets with much fewer parameters and MAdd than ResNet (He et al., 2016) and VGGNet (Simonyan & Zisserman, 2014). For a fair comparison, we follow the experiment settings as in the original papers.

**Results on ImageNet.** We first conduct experiments on the ImageNet dataset to investigate the effectiveness of our method. We use the same width multiplier as in Sandler et al. (2018) as our baseline. We choose four common width multiplier values, *i.e.*, 0.75, 0.5, 0.35 and 0.18 (Sandler et al., 2018; Zhang et al., 2018; Yu et al., 2018) for a fair comparison with other compression approaches.

The performance of our method and the baseline is summarized in Table 2. For all the width multiplier values, our method outperforms the baseline by a significant margin. It can be observed that under higher compression ratio the performance gain is also larger. Moreover, the performance of MobileNetV2 drops significantly when the compression ratio is larger than $3\times$. However, our NES method increases the performance by 3.78% at a large compression ratio. This is because when the

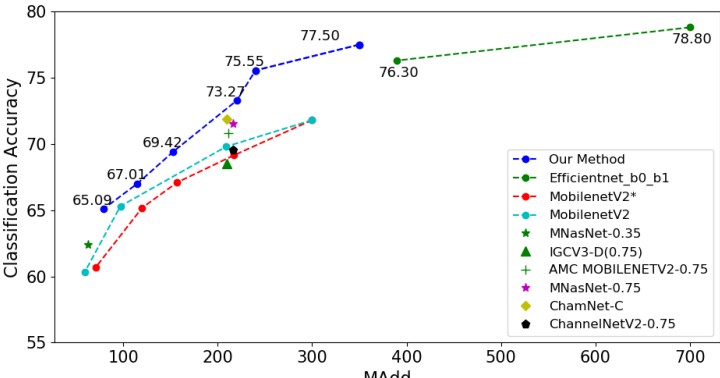

Figure 5: ImageNet classification accuracy of our method, EfficientNet, MobileNetV2 baselines and other NAS based methods including AMC (He et al., 2018), IGCV3 (Sun et al., 2018), MNasNet (Tan et al., 2018), ChamNet (Dai et al., 2018) and ChannelNet (Gao et al., 2018). Our method outperforms all the methods within the same level of MAdd. Here, MobileNetV2* is our implementation of baseline models and MobileNetV2 is the original model with width multiplier of 0.35,0,5,0,75 and 1. The backbone model for our results are EfficientNet-b1 and b0 (Tan & Le, 2019) with multiplier 0.5 and MobileNetV2 with multiplier of 0.75, 0.5, 0.35 and 0.2, respectively (from top to bottom). Note that we do not use additional training tricks including the squeeze-and-excitation module (Hu et al., 2018) and the Swish activation function (Ramachandran et al., 2017).

compression ratio is high, each layer in the baseline model does not have enough capacity to learn good representations. Our method is able to generate more expressive weights from the epitome with a learned transformation function.

We also compare our method with the state-of-the-art compression methods in Table 3. Since an optimized architecture tends to allocate more channels to upper layers (He et al., 2018; Yu & Huang, 2019a), we also run experiments with larger size of the epitome for upper layers. The results is denoted with suffix '-A' in Table 3. Comparison with more models are shown in Figure 5. As shown, NES outperforms the current SOTA mobile model (less than 400M MAdd model) EfficientNet-b0 by 1.2% with 40M less MAdd. Obviously, our method performs even better than some NAS-based methods. Although our method does not modify the model architecture, the transformation from the epitome to the convolution kernel optimizes the parameter allocation and enriches the model capacity through the learned weight combination and sharing.

**Results on CIFAR-10.** We also conduct experiments on CIFAR-10 dataset to verify the efficiency of our method as shown in Table 6. Our method achieves $3.5\times$ MAdd reduction and $5.64\times$ model size reduction with only 1% accuracy drop, outperforming NAS-based AUTO-SLIM (Yu & Huang, 2019a). More experiments and implementation details are shown in the supplementary material.

**Discussion.** From the above results, one can observe significant improvements of our method over competitive baselines, even the latest architecture search based methods. The improvement of our method mainly comes from alleviating the performance degradation due to insufficient model size by learning richer and more reasonable combination of weights parameters and allocating suitable weight parameters sharing among different filters. The learned transformation from the epitome to the convolution kernel increases the weight representation capability with less increase on the memory footprint and the computation cost. This distinguishes our method from previous compression methods and improves the model performance significantly under the same compression ratio.

## 5 CONCLUSION

We present a novel neural epitome search method which can reuse the parameters efficiently to reduce the model size and MAdd with minimum classification accuracy drop or even increased accuracy in certain cases. Motivated by the observation that the parameters can be disentangled form the architecture, we propose a novel method to learn the transformation rule between the filters to make the transformation adaptive to the dataset of interest. We demonstrate the effectiveness of the method on CIFAR-10 and ImageNet dataset with extensive experimental results.

**Acknowledgement** Jiashi Feng was partially supported by NUS IDS R-263-000-C67-646, ECRA R-263-000-C87-133, MOE Tier-II R-263-000-D17-112 and AI.SG R-263-000-D97-490.

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

# A IMPLEMENTATION DETAILS

## A.1 MOBILENETV2 SETTINGS

**Epitome dimensions for MobileNetV2 bottleneck.** With our NES method, the shape of the feature map produced by each layer can be kept the same as the ones from the original model before compression. However, the number of channels in the feature map is reduced using the width multiplier method for MobileNetV2. Hence, for a fair comparison, we only apply the width multiplier on the output dimension of the first $1 \times 1$ convolutional layer and the input channel dimension of the second $1 \times 1$ convolutional layer within the bottleneck blocks of MobileNetV2 for obtaining the same feature map shape between blocks as our method. Based on this principle, we generate weight tensor based on the epitome along the filter dimension for the first $1 \times 1$ convolutional layers within the bottleneck and along the input channel dimension for the second $1 \times 1$ convolutional layers.

Specifically, we set the epitome shape per layer as $(\#\text{in\_channels}, \frac{\#\text{out\_channels} \times \text{expansion}}{\text{multiplier}}, 1, k)$ for the first $1 \times 1$ convolution layer and $(\frac{\#\text{in\_channels} \times expansion}{\text{multiplier}}, \#\text{out\_channels}, 1, k)$ for the second $1 \times 1$ layer as shown in Table 4. Here, expansion is referring to the ratio between the input size of the bottleneck and the inner size as detailed in Figure 2 of (Sandler et al., 2018). The shape represents the number of input channels, the number of output channels and the kernel size, respectively. The compression ratio for each layer, $c$, can thus be calculated as $c = \frac{1}{multiplier}$.

Table 4: Epitome dimensions for the inverted residual blocks of the MobileNetV2 backbone. Here $w, h, k, k'$ denotes the spatial size, input channels and output channels of the input feature map respectively. Variables $w_c$ and $h_c$ denote the spatial size of the epitome and are set to 1 for $1 \times 1$ convolutional layer. $c$ is used to set the compression ratio for each layer and is similar to the concept of width multiplier as defined in MobileNet (Howard et al., 2017). $t$ is the expansion ratio as defined in MobileNetV2.

| Input | Operators | Output | Epitome | Comp. ratio |
|---|---|---|---|---|
| $h \times w \times k$ | $1 \times 1$, conv2d, ReLU6 | $h \times w \times tk$ | $w_c \times h_c \times k \times ctk$ | $\frac{w_c \times h_c}{c}$ |
| $\frac{h}{s} \times \frac{w}{s} \times tk$ | $3 \times 3$, depth-wise separable, ReLU6 | $\frac{h}{s} \times \frac{w}{s} \times tk$ | $-$ | $1$ |
| $\frac{h}{s} \times \frac{w}{s} \times tk$ | $1 \times 1$, conv2d, linear | $\frac{h}{s} \times \frac{w}{s} \times k'$ | $w_c \times h_c \times ctk \times k'$ | $\frac{w_c \times h_c}{c}$ |

## A.2 EPITOME DIMENSION DESIGN

The size of the epitome can be calculated precisely by the original model and the desired compression ratio $r$. For a CNN with $C$ $n$-dimensional convolutional layers and $K$ fully-connected layers, its number of parameters can be calculated as $\sum_{i=1}^{C} \prod_d L_d^i + \sum_{k=1}^{K} N_{in}^k N_{out}^k$, where $L_d^i$ denotes the length of the convolution weight tensor along the $d^{th}$ dimension of the $i^{th}$ convolutional layer. $N_{in}^k$ and $N_{out}^k$ denote the input and output dimension of the $k^{th}$ fully-connected layer.

We assign an epitome $E^j \in \mathbb{R}^{W_j^E \times H_j^E}$ for each layer $j$, and a routing map $\mathcal{M} : (x_1, x_2, \ldots, x_n) \to (p, q)$, i.e., the weight value of an $n$-d filter at location $(x_1, x_2, \ldots, x_n)$ being equal to $E(p, q)$. We define the dimension of the epitome to be 2D here to illustrate the general case and later, we will show that in practice, the dimension of the epitome can be increased to save the computation memory. The size of the total epitome is, thus, $\sum_{j=1}^{C+K} W_j^E \times H_j^E$. After learning the routing map for layer $j$, we store the location mapping as a lookup table of size $M_j$. The size of all the mapping tables is $\sum_{j=1}^{C+K} M_j$. Hence, the compression ratio can be calculated as

$$r = \frac{\sum_i^C \prod_d L_i^d + \sum_k^K N_{in}^k N_{out}^k}{\sum_j^{C+K} (W_j^E \times H_j^E + M_j)}. \tag{16}$$

In our analysis, we use a uniform compression ratio for all the layers. Therefore, given a compression ratio $r$, the size of the epitome for each layer can be calculated accordingly. This deterministic design of the epitome size is hardware friendly and can be used to control the memory allocation.

**Epitome patch design** We set the patch size along each dimension to be $w, h, \beta_1$ and $\beta_2$. Note that the transformation along each dimension are independent and hence can be conducted separately. The starting index of the transformation along the spatial dimension and the input channel dimension are learned in pairs. This is because the transformation along the spatial dimension will also increase the input channel dimension.

### A.3 INDEX SEARCH SPACE IN EPITOME

The selection space of the starting index for the transformation is not all the indices available along the channel dimension. We partition the channels into groups to build a super-index with a group length $l_g$. For example, for an epitome that has C channels along the input channel dimension, the potential channel index ranges from 0 to C - 1. With our super-index scheme, adjacent $l_g$ channels are grouped as a single index $C_g$ and thus, the selection space of the index ranges from 0 to $C/l_g - 1$. The range of the super-index is used to scale the output from the transformation learner.

## B MORE RESULTS ON 1D CONVOLUTION COMPRESSION

We also conduct experiments to examine the highest compression ratio that our method can achieve without performance drop compared to WSNet (Jin et al., 2017). For a fair comparison, we choose the same 8-layer CNN model backbone as used in WSNet. Configuration details have been demonstrated in Table 1 in the formal paper. The results are shown in Table 5.

Table 5: Comparison with WSNet on ESC-50 dataset. We choose the compressed model by WSNet method as our baseline. By decreasing the size of the epitome, we can achieve higher compression ratio. We apply uniform compression ratio for all layers. It is observed that the highest compression ratio we can achieve before our method's performance become smaller than WSNet is $3.16\times$.

| Methods | Compression Rate | Accuracy (%) (top1) |
|---|---|---|
| WSNet | $1.00\times$ | 66.5 |
| Our method-1 | $1\times$ | 73.0 |
| Our method-2 | $2.35\times$ | 69.25 |
| Our method-3 | $3.16\times$ | 65.5 |

## C EXPERIMENT RESULTS ON CIFAR10

The experiment results are shown in Table 6

Table 6: Comparison with other state-of-the-art models in classification accuracy on CIFAR-10. Our method outperforms the recently proposed AUTO-SLIM (Yu & Huang, 2019a) which is a compression method using AutoML method.

| Methods | Parameters | MAdd (M) | Top-1 Accuracy(%) |
|---|---|---|---|
| MobilenetV2-1 | 2.2M | 93.52 | 94.06[*] |
| MobilenetV2-0.18[*] | 0.4 M | 21.59 | 91.70 |
| Auto-Slim | 0.7M | 59 | 93.00 |
| Auto-Slim | 0.3M | 28 | 92.00 |
| Our method | 0.39M | 26.8 | $\mathbf{93.22 \pm 0.013}$ |

## D NES FOR FULLY-CONNECTED LAYER

Let $D \in \mathbb{R}^{N_{in} \times N_{out}}$ denotes the parameter matrix for a fully-connected (FC) layer. We could use Eqn. (18) to sampling along the input dimension and Eqn. (23) to sample along the output dimension. We also conduct experiments on CIFAR-10 dataset to verify the efficacy of our method on FC layers as shown in Table 7. We take MobileNetV2-1.0 as our baseline.

Table 7: Experiment results of our method applied on fully connected layer of MobileNetV2.

| Methods | Parameters | Model Comp. Rate | FC Param. Comp. Rate | Top-1 Acc. (%) |
|---|---|---|---|---|
| MobileNetV2-1 | 2.2M | $1.00\times$ | $1.00\times$ | 94.06 |
| MobileNetV2-0.5 FC[*] | 0.4 M | $5.55\times$ | $2.00\times$ | 91.8 |
| Our method-0.5 FC | 0.39M | $5.64\times$ | $2.00\times$ | **92.96** |

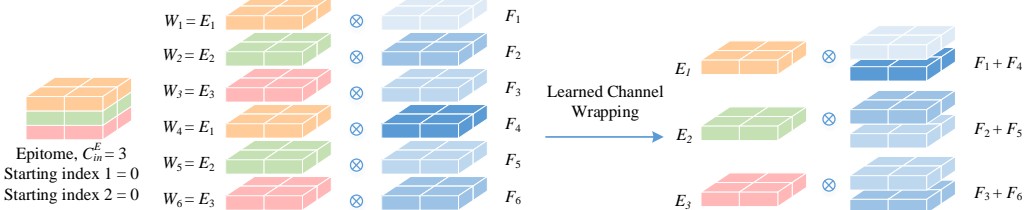

Figure 6: NES transformation along the input channel dimension with channel wrapping. To simplify the illustration, we choose an epitome with shape $\mathbb{R}^{w \times h \times 3 \times 1}$ and $\beta_1 = 3$. The transformation is applied twice and the two starting indices are 0. The input feature map $\mathcal{F}$ is first grouped based on the learned interpolation position of the kernels. Input feature map $\mathcal{F}_1$ and $\mathcal{F}_4$ are both multiplied with the first channel in the epitome since $W_1$ and $W_4$ are both generated with $E_1$. To reuse the multiplication, feature map $\mathcal{F}_1$ and $\mathcal{F}_4$ are first added together before multiplying with the weights kernel $E_1$. The figure uses integer index to simplify the illustration. When the learned indices are fractions, the feature maps are the weighted summation of the two nearest integer indexed sub-tensors in the epitome as shown in Eqn. (7). For example, if the two starting indices in this figure are 0.6 and 0.3, the calculation becomes $(0.6F_1 + 0.3F_4 + 0.4F_3 + 0.7F_6) \otimes E1 + (0.4F_1 + 0.7F_4 + 0.6F_2 + 0.3F_5) \otimes E_2 + (0.4F_2 + 0.7F_5 + 0.6F_3 + 0.3F_6) \otimes E_3$. Since we group the feature map first before the convolution, the cost is reduced.

# E  PROOF ON PARAMETER AND COMPUTATION REDUCTION

With the epitome $E$ as defined in the main text, our method introduces a novel transformation function where the convolution filter weights are transformed from $E$ with $\tau(\cdot)$. In this section, we start with the most general situation where the epitome is two-dimensional, $E \in \mathbb{R}^{W^E \times H^E}$, and we will show that increasing the dimension of epitome can reduce the computation memory cost aggressively. Our method can be extended to $n$-dimension convolution and fully-connected layers straightforwardly. The transformation process along the input channel dimension is illustrated in Figure 3 In our method, all the weight parameters $\theta_{i,j,m,c}$ are transformed from the compact epitome $E$. The convolution with NES is shown as below:

$$G_{t_w,t_h,c} = f_{t_w,t_h} * k_c = \sum_i^W \sum_j^H \sum_m^{C_{in}} F_{t_w+i,t_h+j,m} E_{\mathcal{M}(i,j,m,c)}. \tag{17}$$

**Proof on computational cost reduction.** Since the weight elements per convolutional layer are formed based on the same $E$, there is computational redundancy when two convolution kernels are selected from the same portion of $E$ as shown in Figure 2. To reuse the multiplication in convolution, we first compute the multiplication between each element in the epitome $E$ and the input feature map. The results are saved as a *product map* $P$ such that the computations are done only once. However, given the epitome $E \in \mathbb{R}^{W^E \times H^E}$, the product map size is $W \times H \times C_{in} \times W^E \times H^E \times C_{out}$ which consumes large computational memory. To reduce its size, we propose to increase the dimensions of the epitome to $E \in \mathbb{R}^{W^E \times H^E \times C_{in}^E \times C_{out}^E}$ in order to group the computation results in the product map. Each entry in the product map $P$ is calculated as the dot product between the channel dimension along the input feature map and the third dimension along the compact weight matrix. We set $C_{in}^E$ to be smaller than $C_{in}$ to further boost the compression and $\beta_1$ equal to $C_{in}^E$. As illustrated in Figure 4, the input channels of the feature map is first grouped by

$$\tilde{F}(i,j,m) = \sum_{c'=0}^{R_{cin}-1} F(p,q,m+c' \times C_{in}^E + c_{in}), \tag{18}$$

where $R_{cin} = \lceil C_{in}/C_{in}^E \rceil$ is the compression ratio along the input channel dimension and $m + c_{in} + c \times C_{in}^E$ is the learned position with $(p,q,c_{in},n) = \mathcal{M}(i,j,c,m)$, where $c_{in} \in [1, C_{in}^E]$. The transformed input feature map $\tilde{F}$ has the same number of channels as $C_{in}^E$. Let $(i,j,m)$ index the transformed feature map location. The product map is then calculated as

$$P_{i,j,p,q,n} = \tilde{\mathcal{F}}_{i,j,:} \cdot E'_{p,q,:,n}, \tag{19}$$

where $\cdot$ denotes the dot product operation.

Now, the multiplications can be reused by replacing the convolution kernels with the product map $P$. Based on Eqns. (17), (18), (19), the convolution can be reduced by

$$G_{t_w,t_h,n} = \sum_{i=0}^{W-1} \sum_{j=0}^{H-1} P_{(t_w+i,t_h+j,p,q,n)}. \tag{20}$$

To reuse the additions, we adopt an integral image $I$ which is proposed in (Crow, 1984) but differently we extend the integral image dimension to make it suitable for 2D convolution based on $P$. Our integral image can be constructed by

$$I(t_w, t_h, p, q, n) = \begin{cases} P_{0,t_h,p,q,n}, & t_w = 0 \\ P_{t_w,0,p,q,n}, & t_h = 0 \\ P_{t_w,t_h,0,q,n}, & p = 0 \\ P_{t_w,t_h,p,0,n}, & q = 0 \\ P_{t_w,t_h,p,q,0}, & n = 0 \\ I(t_w-1, t_h-1, p-1, q-1, n-1) + P(t_w, t_h, p, q, n), & \text{else.} \end{cases} \quad (21)$$

From Eqn. (21), the 2D convolution results can be retrieved in a similar way to (Jin et al., 2017) as follows:

$$G_{t_w,t_h,p,q,n} = I(t_w+w-1, t_h+h-1, p+w-1, q+h-1, n) - I(t_w-1, t_h-1, p-1, q-1, n-1). \quad (22)$$

As we set $C_{out}^E$ to be smaller than the output channel dimension of the convolution kernel, we reuse the computation results from the epitome via

$$\tilde{G}_{t_w,t_h,r_{out}\times\beta_2:(r_{out}+1)\times\beta_2} = G_{t_w,t_h,n:n+\beta_2}, \quad (23)$$

where $r_{out} \in \{0, 1, ..., R_{cout} - 1\}$, $R_{cout} = \lceil C_{out}/\beta_2 \rceil$ is the number of samplings conducted along the output channel dimension, and $n = \mathcal{M}(r_{out} \times \beta_2)$ is the learned mapping along the filter dimension. The filter length $\beta_2 \in \{1, 2, ..., C_{out}^E\}$ is a hyper-parameter and is decided empirically. In our experiments, we choose $\beta_2 = C_{out}$. Thus, the MAdds can be calculated as

$$\text{Reduced MAdd} = \underbrace{(2C_{in}W^E H^E - 1)WHC_{out}^E}_{\text{From Eqn. (19)}} + \underbrace{WHW^E H^E C_{out}^E}_{\text{From Eqn. (21)}} + \underbrace{2R_{cin}WH\beta_1}_{\text{From Eqn. (18)}} + \underbrace{2R_{cout}\beta_2}_{\text{From Eqn.(23)}}. \quad (24)$$

Suppose we use sliding window with a stride of 1 and no bias term, the MAdds of the conventional convolution can be calculated based on Eqn. (3) as shown below:

$$\text{MAdd} = (2 \times C_{in} \times w \times h - 1) \times H \times W \times C_{out}. \quad (25)$$

Therefore, the total MAdd reduction ratio is

$$\text{MAdd Reduction Ratio} = \frac{C_{out}HW(2C_{in}wh - 1)}{C_{out}^E HW(W^E H^E + 2C_{in}^E W^E H^E - 1) + 2R_{cin}WH\beta_1 + 2\beta_2 R_{cout}} \\ \approx \frac{C_{out}C_{in}wh}{C_{out}^E C_{in}^E W^E H^E} \quad (26)$$

**Proof on parameter reduction.** The parameter compression ratio for a 2D convolution layer can be calculated as follows:

$$r = \frac{whC_{in}C_{out}}{W^E \times H^E \times C_{in}^E \times C_{out}^E + 3 \times R_{cin} + R_{cout}}, \quad (27)$$

where $w$ and $h$ denote the width and height of the kernel of the layer. $W_c$ and $H_c$ denote the spatial size of the corresponding epitome. From Eqn. (26) and Eqn. (27), it can be observed that the compression ratio is mainly decided by $\frac{C_{out}C_{in}wh}{C_{out}^E C_{in}^E W^E H^E}$.

