# OpenReview forum: "Neural Epitome Search for Architecture-Agnostic Network Compression"
_ICLR.cc/2020/Conference — Accept (Poster)_

### Official Review · AnonReviewer1 · 2019-10-15
**Official Blind Review #1**

**Rating:** 3

**Review:**

In this paper, the authors learn epitomes, which are small weight tensors which can be used with a learnt transform to produce tensors of an appropriate size (e.g. the sizes used in MobileNet v2).  This gives a reduction in the number of parameters required, and the number of MAdds in theory.

This paper is badly written, and could do with a rewrite:

- Citations are used incorrectly (\cite should be used when the citation is meant to be read as part of the sentence).
- "less elements" --> "fewer elements"
- "misuse the notion" --> "abuse the notation"?
- "for fair comparisons" --> "for a fair comparison"

The method is poorly explained; I have read Section 3.2 several times, and I'm still not entirely certain of what's going on. Figure 1 is helpful, but Figure 2 is not, and could be redesigned. 2(b) makes it look like you are going from a 3x3 epitome to a 2x2 kernel, which is clearly not what is happening. I think it would be helpful to give a detailed pictoral example of an epitome mapping to a weight tensor, with arrows between relevant indices changing.

On initial reading, I thought the method allowed dynamic allocation of your epitomes to lots of different tensor sizes. From what I can tell, the network has to be trained from scratch for each possible size, so it isn't flexible in that respect.  From what I can gather, the paper is presenting an alternate approach to *downscale* networks, as opposed to say, reducing width or depth. The comparisons to different widths of MobileNet v2 make more sense under this scenario.

The main sell of the methods appears to be on the basis of MAdd reduction. This makes me nervous, as it doesn't necessarily correspond to actual speed-up or a reduction in energy (see https://arxiv.org/abs/1801.04326). EfficientNet was mainly about Madds too, but they provided some inference times. Would it be possible to add these? The method used with the integral image sounds expensive.

The results look good, but error bars,on the CIFAR experiments at the very least, would be appreciated.

Pros:
-------
- Good results
- Method appears largely novel (although bears some resemblance to https://arxiv.org/abs/1906.04309)

Cons:
--------
- Badly written
- The method is poorly explained
- Uncertainty re: MAdds as a primary comparator

I propose a weak reject for this paper for two primary reasons:
1) The standard of writing, and the explanation of the all-important method are not up to scratch for a top tier conference
2) I have concerns regarding the MAdd calculations. Perhaps you could provide some pseudo-code in the author response?

I am happy to upgrade my score if the authors deal with these issues sufficiently.

**Experience Assessment:**

I have published one or two papers in this area.

**Review Assessment: Checking Correctness Of Derivations And Theory:**

I did not assess the derivations or theory.

**Review Assessment: Checking Correctness Of Experiments:**

I assessed the sensibility of the experiments.

**Review Assessment: Thoroughness In Paper Reading:**

I read the paper at least twice and used my best judgement in assessing the paper.

---

> ### Author Response · Authors · 2019-11-15
> **Response to AnonReviewer3 comments and suggestions (3/3)**
>
> The pseudo code for the madd calculation is shown as below:
>
> Pseudo code for Madd calculation:
>
> def conv_flops_counter_hook(conv_module, input, output):
> # calculation based on equation 21 in the paper
>     batch_size = input.shape[0]
>     output_dims = list(output.shape[2:])
>
>     kernel_dims = list(conv_module.kernel_size)
>     in_channels = conv_module.in_channels
>     out_channels = conv_module.out_channels
>     epi_in = self.weight.shape[1]
>     epi_out = self.weight.shape[0]
>     epi_w, epi_h = self.weight.shape[2], self.weight.shape[3]
>
>     # we choose \beta_1 = epi_in and \beta_2 = epi_out empirically
>     rep_dim1 = out_channels // epi_in
>     rep_dim0 = in_channels // epi_out
>
>     filters_per_channel = out_channels // groups
>     product_map_cost = (2* in_channels * epi_w * epi_h -1) * np.prod(output_dims) * epi_out
>     integral_map_cost = np.prod(output_dims) * epi_w * epi_h * epi_out
>     channel_trans_cost = 2 * rep_dim1 * np.prod(output_dims)
>     filter_trans_cost = 2 * rep_dim0
>
>     conv_per_position_flops = product_map_cost  + integral_map_cost  +
>                                                channel_trans_cost  + filter_trans_cost

---

> ### Author Response · Authors · 2019-11-15
> **Response to AnonReviewer3 comments and suggestions (2/3)**
>
> Madd calculation
>
> •	There are fours parts that contributes to the computation cost: (1) the convolution between the epitome and the input feature map to construct the product map; (2) the construction of the integral map; (3) channel wrapping along the input channel dimension and (4) transformation along the filter dimension.
> •	The computation cost for the product map comes from the convolution between the input feature map and the epitome and hence it is calculated as: 2 * (C_{in} * W^{E} * H^{E} − 1 )W * H * C^{E}_{out}
> •	The computation cost of the integral map is the addition used to construct the map. The number of additions is proportional to the number of elements in the map and hence it is calculated as: W * H * W^{E} * H^{E} * C^{E}_{out}
> •	The computation cost for the transformation along the input channel dimension comes from the channel wrapping. It is equal to the number of additions between the feature map and multiplications with the coefficient as introduced in Eqn.(11) in the main text. Thus, the total computation is: 2Rcin * W * H * β_{1}
> •	The computation cost of the transformation along the filter dimension is similar to the one along the input channel dimensions. The only difference is that channel wrapping is not needed and hence the total computation cost is computed as: 2R_{cout} * β_{2}
> •	The total computation cost is thus calculated as: 2 * C_{in}W^{E} 8 H^{E} − 1 )W * H * C^{E}_{out} + W * H * W^{E} * H^{E} * C^{E}_{out} +  2Rcin * W * H * β_{1} + 2R_{cout} * β_{2}
> •	The size of the integral map can be quite small in the implementation by setting the spatial dimension of the epitome to be small.
> •	We have added in more details in section 3.4 regarding how the computation can be saved. Madd is calculated theoretically. The actually speed up is not fully proportionally to the reduction in madd. However, we believe those engineering works can be done with optimized coding and driver layer.
> •	The calculation of Madd is based on equation 12 and the detailed derivation can be found  in Appendix D in the revised version of the paper.
> •	The pseudo code for Madd calculation is shown at the end of the reply
>
> Difference between NES and Associative Convolutional Layers(ACL):
>
> Both NES and ACL targets to express the convolution layers with a compact set. ACL builds on the basis of tensor low-rank factorization. It first divides a tensor into a set of basises (or the so called slice in the paper) and then uses a convolution slice generator to form the weight tensor used for convolution. The generator  and code word used in ACL can be regarded as a fully connected layer. The core idea is similar to decompose a convolution layer into a set of smaller convolution tensors followed by a fully connected layer. Differently, our method uses an index learner to learn the starting index of a sub-tensor and use a bi-linear interpolation based method to generate the weight tensor with much larger dimensions. The advantages of our method over ACL are summarized as below:
>
> •	Our method has much less storage and computation overhead than ACL method. The slice generator and the code word used in ACL can be regarded as a fully connected layer to combine all slices. The generator itself consumes the storage memory. Besides, using a fully connected layer to combine all slice basis might consume a large computation memory and the computation cost is not reported in the paper. Differently, our method removes the index learner by implementing a routing map. Both the memory and computation cost can be saved using our method.
> •	In ACL, the partition of the weight tensor into slice is a manual work. When applied on different backbone networks, the partition scheme might need to be re-designed. However, in our method, we use an index learner to learn the starting index and then generate the weight tensor based on a non-parametric interpolation function. No manual work is needed for the generation process.
> •	The experiments in ACL focus on Densenet and ResNet. ResNet is popular but redundancy is higher than the current SOTA mobile models. Densenet is efficient on memory but the computation cost is expensive and the computation cost is not reported in ACL. Differently, our experiment focuses on current SOTA mobile networks such as MobileNetv2 and EfficientNet and we report both the memory size and computation results.
> •	The key idea of this paper is to treat the network architecture and parameters separately and search a transformation function that can match the gap between the architecture and the parameter. This is expected to give a new search space for neural architecture design.

---

> ### Author Response · Authors · 2019-11-15
> **Response to AnonReviewer1 comments and suggestions  (1/3)**
>
> Thank you for your detailed review! Your questions are valuable, and we are happy to address your concerns.
>
> Reply to the writing of the paper
>
> •	We have revised the paper and correct the citation errors and revise the words carefully
> •	We have re-organized section 3 to improve the presentation of the explanations of the method.  We have re-designed figure 2 to give a detailed illustration of how the weight tensor can be generated using the epitome along the spatial dimension. We have also added Figure 3  to illustrate the transformation process along the input channel dimension. The transformation along the filter dimension is the same as the transformation along the input channel dimension.
>
> Reply to the potential extension to dynamic allocation:
>
> •	For each designed scheme, the network needs to be trained from scratch. We do not try the dynamic allocation experiments so far. But it is interesting and useful if a single epitome can be used to construct different networks without re-training. Our method can be extended intuitively to achieve the target by incorporating the concept of switch introduced in the slimmable neural network [1]. More specifically, we can put in four switches where each switch controls the percentage of epitome that can be used for the transformation and we can learn four different transformation rules for each switch. In that case, our epitome in each layer can be used to generate different weight tensors dynamically. However, we put it as a separate work from the current version since it is an extension to the current method and can always be integrated into the current method.
>
> Reply to the experiments
>
> Latency measurement
> •	Initially, we use Madd, number of parameters and the classification accuracy as metrics for comparisons with other methods. We do not measure the energy due to the hardware limitation. We have added in one more experiments to measure the latency of our method and the results are shown as below:
>
> Method		                  Latency	Device		Batch Size	 Madd(M)
> ----------------------------------------------------------------------------------------------------
> MBV2-0.5		           26 ms	         CPU		       1		   100
> MBV2-0.75		           28 ms	         CPU		       1		   220
> MBV2-1.0		           34 ms	         CPU		       1		   310
> MBV2-1.4		           42 ms	         CPU		       1		   590
> MBV2-1.4		           7.79 s	         CPU		   1000		   590
> ----------------------------------------------------------------------------------------------------
> MBV2-1.0 + NES-0.5	   27 ms	         CPU	               1                 157
> MBV2-1.4 + NES-0.5	   33 ms	         CPU	               1                 340
> MBV2-1.4 + NES-0.5	   6.27 s	         CPU                   1000  	          340
>
> The server that we have used for the evaluation has 80 Intel(R) Xeon(R) E5-2698 v4 @ 2.20GHz CPUs. As CPU adjusts its power automatically, we put two dry run to change the CPU to maximum power mode. We set the batch size to 1 and run 1000 iterations. We first use MobileNetV2 as a baseline to check the latency measurement. We found that the effect of the speedup is larger when the base model’s computation is large. Hence, we increase the computation by using a MBV2-1.4 base model with 1000 batchsize. To have more accurate measurement, engineering works on the hardware setup is needed. We will try to add in more latency measurements using mobile devices later.
>
> Error bar on CIFAR-10 dataset
>
> •	We re-train our methods on CIFAR-10 dataset for 8 times and calculate the statistics as shown in the table below:
>
> Method		            Params(M)	      Madd(M)		     Accuracy(%)
> -----------------------------------------------------------------------------------------------------------
> MBV2-1.0		         2.2  	                CPU	                   94.06
> MBV2-0.18		         0.4 	                        CPU		           91.7
> -----------------------------------------------------------------------------------------------------------
> MBV2-1.0 + NES-0.2	 0.39 	                CPU	               93.19± 0.013
>
>
> [1] Yu, Jiahui, et al. "Slimmable neural networks." arXiv preprint arXiv:1812.08928 (2018).

---

### Official Review · AnonReviewer2 · 2019-10-26
**Official Blind Review #2**

**Rating:** 6

**Review:**

In this work, the authors describe a technique for compressing neural networks by learning a so-called Epitome (E), and a transformation function (\theta) such that the weights for each layer can be constructed using \theta(E). The epitome and the transformation function can be learnt jointly while optimizing the network for the task specific loss.

The main idea of this paper is really interesting, and the experimental results which compare against other recent techniques also validate the proposed technique. However, while I think the main idea is relatively clear, I personally found the description of the proposed techniques -- particularly 3.2 and 3.4 -- to be somewhat hard to follow, particularly given that some details only appear in the Appendix. I would suggest that the authors try to revise this section by trying to move Figures 4 and 5 from the appendix into the main text. Some additional suggestions also appear below.

Overall, I would while I like the ideas in this paper, based on the current presentation I am inclined to rate the paper as a “weak reject”, though I would raise my rating if the paper was revised to improve the presentation.

Main comments:
1. The authors mention that “During inference, (the) routing map enables the model to reuse computations when the expanded weight tensors are formed based on the same set of elements in the epitomes and therefore effectively reduces the computation cost.” It would be nice to include some results which indicate what the savings are with the proposed routing map.

2. Section 3.2: There were a few aspects of section 3.2 that I think could be improved for clarity.
A.) Personally, I found Figure 2 somewhat tricky to follow. I would suggest removing the 3x2 “Epitome” and “Generated Kernel” figures on the extreme right of the image, since I’m assuming the only goal of these is to indicate that “orange” and “blue” correspond to “Epitome” and “Kernel” respectively. I would also suggest mentioning the correspondence of the colors as the first sentence in the caption. Finally, if possible, I would suggest adding a small description alongside the (a), (b), (c) subcaptions: e.g., (a) straightforward but non-differentiable mapping, … , (c) Generated Weigh Kernel.

B.) I believe that the authors use a separate E for each layer, and that Epitomes are not shared across layers. I may have missed this in the text, but it would be useful to clarify this explicitly again in the section.

C.) The “parameterized transformation layer” is mentioned before Equation 3. I think it would be useful to mention that this is implemented using neural networks in your work for clarity. E.g.: “To handle the above two obstacles, ... three
parts: (1) a parameterized transformation learner η (implemented using a neural network in this work) used to learn a set of starting indices for patches in epitome (i.e. all elements in the same epitome patch share identical starting indices); … and an interpolation based generator (Eqn. 3).”
or
“To handle the above two obstacles, ... three
parts: (1) a parameterized transformation learner η (See Section 3.3) used to learn a set of starting indices for patches in epitome (i.e. all elements in the same epitome patch share identical starting indices); … and an interpolation based generator (Eqn. 3).”

D.) The exact structure of the “parameterized transformation layer” wasn’t exactly clear to me. In 3.3, the authors mention that it consists of “of two convolutional layer, followed by a sigmoid function … takes the feature map of the convolutional layer as input”. Please clarify exactly what is fed in as the input to this network e.g., (input feature map: F, and the indices i,j).

3. I personally also found Section 3.4 which discusses the Computation reduction was also somewhat hard to follow. Some clarification questions: Is the memory/computation cost of the storing/creating the routing map included in the compression calculations? I think it is important to factor these costs when computing the savings achieved by the model. Also, I was unclear on what R_{cin} and R_{cout} are, and why they appear in Equation 5. Could the authors please clarify.

Minor Comments:
1. Abstract: “Traditional compression methods … all assume that network architectures
and parameters should be hardwired.” What does it mean for them to be “hardwired” in this context?
2. Abstract: “Experiments demonstrate that, … with 25% MAdd reduction and AutoML for Model Compression (AMC) by 2.5% with nearly the same compression ratio.” --> “Experiments demonstrate that, … with 25% MAdd reduction, and a 2.5% Madd reduction for AutoML for Model Compression (AMC) with nearly the same compression ratio.”

------- Update after Author Response --------
I would like to thank the authors for their responses and for the updates which strengthen the paper in my view. I have updated my score accordingly.


**Experience Assessment:**

I have published one or two papers in this area.

**Review Assessment: Checking Correctness Of Derivations And Theory:**

I did not assess the derivations or theory.

**Review Assessment: Checking Correctness Of Experiments:**

I assessed the sensibility of the experiments.

**Review Assessment: Thoroughness In Paper Reading:**

I read the paper at least twice and used my best judgement in assessing the paper.

---

> ### Author Response · Authors · 2019-11-12
> **Response to AnonReviewer2 comments and suggestions**
>
> Thank you for your detailed and valuable review! Following your suggestions, we have updated the paper with improved presentation and we're happy to address your concerns. A summary of the modifications is shown as below:
>
> Reply to major comments:
>
> Computation savings with routing map:
>
> •	The benefits brought by the routing map is that the index learner can be discarded during the inference phase, and the outputs of the index learner is recorded in the routing map and saved as a look-up table. With the implementation of the routing map, the computation overhead due to the index learner is removed. We have revised section 3.2 to state this benefit clearer in a separate paragraph named as “Routing map”
>
> Redesign of the weight tensor generation process(figure 2 in original version):
>
> •	We agree that previous figure 2 is not clear on the generation process. We have modified figure 2 in section 3.2 to improve the presentation. We combined figure 4 and figure 2 in the original version to give an example of the transformation process along the spatial dimension with the structure of the index learner. We also added figure 3 to illustrate the transformation along the channel dimensions. Based on figure 3, we put in more details regarding the meaning of R_{cin}. The meaning of R_{cin} is the number of samplings applied along the input channel dimension. The transformation along the filter dimension is same as the transformation along the input channel dimension. The difference is that the computation reuse along the input channel dimension needs special techniques of channel wrapping. We illustrate the channel wrapping process in Figure 4 of the updated version.
>
> Yes, each layer has an epitome and the epitome is not shared among layers. We have revised the paper to emphasize this point in section 3.2 paragraph “Indexing function”.
>
> Design of the transformation function:
>
> •	We use separate transformation learners for each layer. The learner is implemented by  two convolution layers. It takes in the input feature tensor of the corresponding layer and outputs the mapped starting index of the sub-tensor in the epitome. The sub-tensor at the new position will be used to construct the weight tensor for convolution. We also show the structure of the learner in the revised figure 2. We rename this transformation learner as index learner in the revised version in section 3.2.
>
>
> Memory and computation compression calculation of the routing map:
>
> •	The memory cost of the routing map is 3 x R_{cin} + R_{cout}. The routing map is used to record the index mapping between the sub-tensor in the epitome and the weight tensor. This is implemented as a look-up table manner. Because we take a patch of the tensor in the epitome, only the starting indices are recorded. The position of the rest of the elements can be calculated as starting index + offset where offset is the relative position for those elements in the sub-tensor.
>
> Explanation of R_{cin} and R_{cout}:
>
> •	For example, with a learned routing map M, the sub-tensor in the epitome that will be used to fill up the sub-tensor at position (i,j,m) in the weight tensor is E[p:p+w, q:q+h, c’_{in}:c’_{in} + \beta_1, :], where (p,q, c’_{in}) = M(i,j,m). The selected sub-tensors in the epitome are concatenated together to form a larger tensor. Hence, the number of such indices is calculated as C’_{in} / \beta_1 and this is noted as R_{cin}. Since each recorded index has three values, the size of the routing map is 3 x R_{cin}. Similarly,   along the filter dimension, the size is R_{cout} where R_{cout} = C '_{out}/ \beta_2. Here, \beta_1 and \beta_2 denotes the length of the sub-tensor along the input channel dimension and the filter dimension that we select from the epitome. We have put in more details regarding the meaning of R_{cin} and R_{cout} in section 3.2. Since the routing map is implemented as a look-up table, the computation is O(1). The formulation for the computation of the compression ratio has taken the routing map into consideration.
>
> Reply to minor comments:
>
> •	Meaning of “hardwired”: this term is supposed to describe the relationship between the neural network architecture and the corresponding parameters. In this work, we consider the network architecture and the parameters as separate components. The word “hardwired” is used to describe the one-to-one correspondence between the architecture and the parameters. We have updated the abstract to make this clearer.
> •	We have modified the experiments summary part in the abstract as pointed out.

---

### Official Review · AnonReviewer3 · 2019-11-05
**Official Blind Review #3**

**Rating:** 6

**Review:**

This paper focuses on the problem of  neural network compression, and proposes a new scheme, the neural epitome search. It learns to find compact yet expressive epitomes for weight parameters of a specified network architecture. The learned weight tensors are independent of the architecture design.  It can be encapsulated as a drop in replacement to the current
convolutional operator. It can incur less performance drop. Experiments are conducted to show the effectiveness of the proposed method. However, there are some concerns to be addressed.
-It is not too clear how to learn the epitomes and transformation functions.
-Authors stated that the proposed method is independent of the architecture design. From the current statements, it is not explained clearly.

**Experience Assessment:**

I have read many papers in this area.

**Review Assessment: Checking Correctness Of Derivations And Theory:**

I assessed the sensibility of the derivations and theory.

**Review Assessment: Checking Correctness Of Experiments:**

I assessed the sensibility of the experiments.

**Review Assessment: Thoroughness In Paper Reading:**

I read the paper at least twice and used my best judgement in assessing the paper.

---

> ### Author Response · Authors · 2019-11-12
> **Response to AnonReviewer3's comments and suggestions**
>
> Thank you for your review! Your questions are valuable, and we are happy to address your concerns. In summary, we explain below how the epitome parameters are updated and why the method is agnostic to the model architecture.
>
> Q1: It is not too clear how to learn the epitome and transformation functions.
>
> •	Epitome parameter updates: the epitome is defined as a parameter tensor with smaller dimensions. The parameters inside the epitome are updated with standard back-propagation to minimize the training loss, after being transformed to instantiate model parameters. The gradient of each parameter in the epitome is the summation of the gradients of all the elements in the convolution weight tensor where the corresponding epitome element is used. The transformation function is end-to-end differentiable and hence the epitome parameters are updated in an end-to-end manner. The learning procedure for epitome  is introduced in detail in Section 3.3. We have revised the section to make it clearer.
> •	Transformation function: We updated the description in Section 3.2 on transformation functions in the paper. In summary, the transformation function includes three parts: (1) an index learner, (2) a routing map and (3) an interpolation based sampler. The transformation function works  as follows: It uses the index learner to learn a group of indices that map the sub-tensor inside the epitome to the convolution weight tensor. The learned indices and the epitome are then fed into an interpolation based sampler. Each pair of the learned starting index will have a corresponding output from the interpolation based sampler. Those outputs are then concatenated together to constitute the weight tensor. The index learner is implemented by a two-layer module. We use separate index learners for each layer and the learner takes the input feature of the corresponding layer as its input and outputs the learned starting indices. Thus, the learner can be optimized via standard back propagation.
>
> Q2: Independence between the epitome and the architecture
>
> •	In section 3, we show that the model size and the computation are mainly related to the size of the epitome. With our proposed transformation function, the shape of the epitome is independent of the model architecture. This is because the transformation function is able to transform any shape of the epitome to a tensor matching the shape of weight tensor for the architecture.

---

### Decision · Program_Chairs · 2019-12-19

**Decision:**

Accept (Poster)

**Comment:**

The paper proposed a novel way to compress arbitrary networks by learning epitiomes and corresponding transformations of them to reconstruct the original weight tensors. The idea is very interesting and the paper presented good experimental validations of the proposed method on state-of-the-art models and showed good MAdd reduction. The authors also put a lot of efforts addressing the concerns of all the reviewers by improving the presentation of the paper, which although can still be further improved, and adding more explanations and validations on the proposed method. Although there's still concerns on whether the reduction of MAdd really transforms to computation reduction, all the reviewers agreed the paper is interesting and useful and further development of such work would be useful too.